# Global Research on Care-Related Burden and Quality of Life of Informal Caregivers for Older Adults: A Bibliometric Analysis

Aliya Zhylkybekova [1,*], Gulbakit K. Koshmaganbetova [1], Afshin Zare [2], Nadiar M. Mussin [3], Asset A. Kaliyev [3], Shabnam Bakhshalizadeh [4,5], Nurgul Ablakimova [6], Andrej M. Grjibovski [7,8,9,10], Natalya Glushkova [11] and Amin Tamadon [2,12,*]

1   Department of Evidence-Based Medicine and Scientific Management, West Kazakhstan Marat Ospanov Medical University, 030012 Aktobe, Kazakhstan; gulbakitkoshmaganbetova@gmail.com
2   PerciaVista R&D Co., Shiraz 1731, Iran; afshinzareresearch@gmail.com
3   Department of Surgery and Urology No. 2, West Kazakhstan Medical University, 030012 Aktobe, Kazakhstan; nadiar_musin@zkmu.kz (N.M.M.); aset_kaliyev@mail.ru (A.A.K.)
4   Reproductive Development, Murdoch Children's Research Institute, Melbourne, VIC 3052, Australia; bakhshalizadehshabnam@gmail.com
5   Department of Paediatrics, University of Melbourne, Melbourne, VIC 3052, Australia
6   Department of Pharmacology, West Kazakhstan Marat Ospanov Medical University, 030012 Aktobe, Kazakhstan; n.ablakimova@zkmu.kz
7   Central Scientific Research Laboratory, Northern State Medical University, 163069 Arkhangelsk, Russia; andrej.grjibovski@gmail.com
8   Department of Epidemiology and Modern Vaccination Technologies, I.M. Sechenov First Moscow State Medical University, 119048 Moscow, Russia
9   Department of Biology, Ecology and Biotechnology, Northern (Arctic) Federal University, 163002 Arkhangelsk, Russia
10  Department of Health Policy and Management, Al-Farabi Kazakh National University, 050040 Almaty, Kazakhstan
11  Department of Epidemiology, Biostatistics and Evidence Based Medicine, Al-Farabi Kazakh National University, 050040 Almaty, Kazakhstan; glushkovanatalyae@gmail.com
12  Department for Natural Resources, West-Kazakhstan Marat Ospanov Medical University, 030012 Aktobe, Kazakhstan
*   Correspondence: zhylkybekovaa@gmail.com (A.Z.); amintamaddon@yahoo.com (A.T.); Tel.: +7-777-660-8696 (A.Z.); +7-705-629-9350 (A.T.)

**Abstract:** As global populations continue to undergo demographic aging, the role of caregivers in providing essential support and assistance to older adults has become increasingly prominent. This demographic shift has led to a growing reliance on informal caregivers, often family members, who take on the responsibilities of caring for older adults. This not only affects immediate family dynamics but also holds broader implications for societal sustainability. The primary objective of this bibliometric analysis is to comprehensively examine the worldwide research output related to the quality of life and caregiver burden among individuals providing care to older adults. By understanding the worldwide research output related to caregivers and their quality of life and burden, we can assess the long-term sustainability of caregiving practices. We retrieved studies with titles containing the terms "caregivers", "burden", "quality of life", and "aged" from the Web of Science (WOS) database. The collected publications were then subjected to analysis using the "bibliometric" package in the R programming environment. A total of 44 publications from 2006–2023 were included in the analysis. Spain emerged as the leading contributor in terms of the number of publications, accounting for 21.9%, followed by the USA at 16.5% and China at 13.6%. The most prolific institution was Kaohsiung Medical University, Taiwan, responsible for 25% of the publications. Among the authors, Cura-Gonzalez I.D. had the highest number of articles, contributing four publications, or 9.1% of the total output. An analysis of co-occurring keywords revealed that the predominant focus of the research revolved around caregiver burden, quality of life, health, care, stress, and impact, reflecting enduring areas of interest within this field. This bibliometric analysis may serve as a tool to provide insights into the current state of research on caregiver burden and quality of life among those caring for older adults. The results of this study can contribute to the

assessment of research strategies and the encouragement of global cooperation in the field of care for older adults. By considering the multidimensional nature of caregiving challenges and promoting international cooperation, strides can be made towards sustainable caregiving practices that ensure the wellbeing of both caregivers and the aging population, thus safeguarding the sustainability of healthcare systems worldwide.

**Keywords:** bibliometric analysis; caregivers; burden; quality of life; aged

## 1. Introduction

As the global population continues to age, the role of caregivers in providing essential support to older adults is becoming increasingly important [1]. The aging demographic landscape results in a greater reliance on informal caregivers, often family members, responsible for the care of older adults [2]. Care provided by family members requires fewer resources compared to institutionalized care, such as nursing homes or assisted living facilities. By utilizing existing household resources, informal caregivers contribute to reduced energy and infrastructure consumption, promoting sustainability. Moreover, informal caregiving offers economic benefits by reducing the financial burden associated with institutionalized care. This can lead to financial savings for the society, indirectly promoting a sustainable economic balance. Informal caregiving can strengthen family bonds and enhance reciprocity among family members. By engaging in reciprocal care relationships, families can support one another through different life stages, including caring for older adults. This sustainable family structure allows for continuous care provision across generations, ensuring the wellbeing of family members without relying solely on external care services.

Informal caregiving has been shown to be associated with several challenges, including the significant burden experienced by caregivers for older adults [3]. Caregiver burden includes physical, emotional, and social stressors arising from continuous care, health management, and the wellbeing of the older adults [4–9]. This burden can affect the quality of life of both the caregivers and the recipients [9].

In the realm of caregiving, "quality of life" includes physical health, psychological wellbeing, social engagement, and overall life satisfaction [10]. Preserving and improving caregivers' quality of life is crucial, given its link to the ability to provide care [9].

Caregivers are susceptible to psychosocial symptoms such as anxiety and depression, with depression being the most prevalent health issue among family caregivers. Caregiving can lead to reduced work productivity, job abandonment, financial difficulties, and a negative impact on job performance. Caregivers also report various health issues, including fatigue, digestive problems, weakened immune systems, slow wound healing, high blood pressure, and sleep disturbances [11]. Social isolation and reduced social connections are common among caregivers, especially those caring for older adults, making them a high-risk group [12]. The World Health Organization (WHO) emphasizes the need to support and ensure caregivers' wellbeing [13].

In the existing academic literature, much research focuses on bibliometric analyses of informal caregivers, mainly those caring for individuals with specific medical conditions like dementia, stroke, and Alzheimer's disease. Most bibliometric analyses indicate dementia as the most extensively studied subject [14–17]. In contrast to the existing literature, which predominantly focuses on specific health conditions such as dementia, stroke, and Alzheimer's disease in bibliometric analyses of informal caregivers, our study represents a more comprehensive examination of the literature. Our analysis extends beyond the confines of individual health conditions. Emphasizing a global perspective, our study considers research from diverse regions and countries, contributing to a better understanding of variations in caregiver burden and quality-of-life experiences on a global scale. By identifying overarching trends, patterns in research publication, collaboration

networks, and impactful journals, our analysis offers insights that transcend the limitations of condition-specific studies.

The aim of this study was to examine global research on the quality of life and caregiver burden for those caring for older adults. By understanding the worldwide research output related to caregivers and their quality of life and burden, we can assess the long-term sustainability of caregiving practices.

## 2. Materials and Methods

### 2.1. Search Strategy

To perform an extensive analysis of research pertaining to the quality of life and caregiver burden experienced by individuals caring for older adults, we collected data from the Web of Science Core Collection (WOS-CC). Our search methodology was designed to be inclusive, encompassing diverse aspects of this field of study. Data collection took place in September 2023 (Appendix A). The search strategy can be summarized as follows: "Caregivers" AND "Burden" AND "Quality of life" AND "Aged" (Title) or "Caregivers" AND "Burden" AND "Quality of life" AND "Aged" (Abstract) (Appendix A). The search covered the period from 2006 to 2023, without time limitations, according to the indexing in the Web of Science database.

We included only original research articles published in English. Other types of publications, papers on other age groups, and papers unrelated to the topics of caregiver burden or quality of life were excluded. The article selection process is demonstrated in Figure 1.

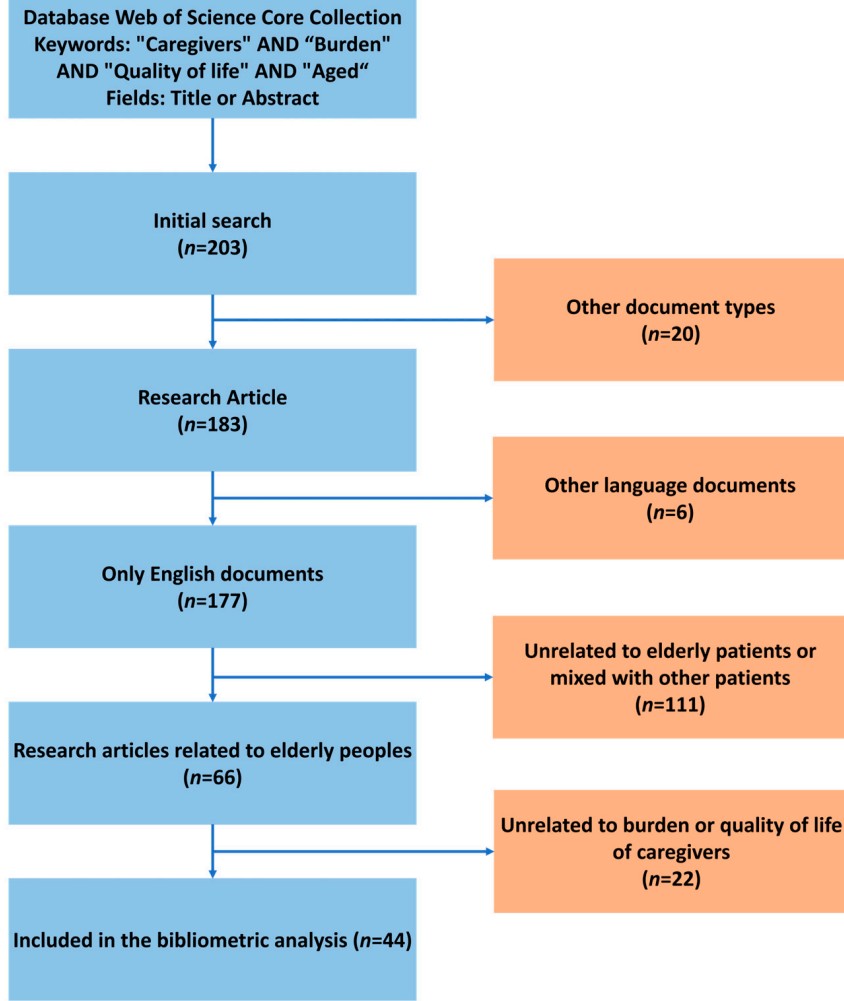

**Figure 1.** Literature selection flowchart.

*2.2. Performance Analysis*

Performance analysis and science mapping were performed using the Bibliometrix R package (http://www.bibliometrix.org; accessed on 23 September 2023) in the R-studio programming environment, version 4.3.1 [18]. The data were analyzed using the Biblioshiny tool. Graphs presented by Biblioshiny were redrawn using GraphPad (GraphPad Prism version 9 for Windows, GraphPad software, San Diego, CA, USA).

We assessed local publication trends and calculated the average total citations per article for each year. The most prolific journals were identified by the number of publications, while the core journals in the field were detected by implementing Bradford's law [19].

*2.3. Identification of Leading Institutions, Sources, Authors, and Collaborating Countries*

We created visual representations of the connections between the most productive institutions and authors to learn more about their cooperation. For productivity, the countries were ranked based on the proportion of papers originating from them. In addition, we measured the extent of cooperation between the 10 most productive countries. To visualize this collaboration, we constructed a map using the number of publications per country.

*2.4. Keyword Frequency Analysis*

We performed a chronological analysis to monitor the occurrence of specific terms throughout different years. To visually depict the distribution and importance of the top 10 most frequently referenced keywords, we created a TreeMap. Furthermore, we conducted a thematic examination to identify the predominant patterns and topics within the selected articles.

## 3. Results

*3.1. Summary of the Papers*

Altogether, we identified 44 relevant studies from 38 different sources published from 2006 to 2023 [20–63]. In total, 314 authors contributed to the abovementioned studies, which received an average of 18.3 citations (SD = 34.9).

Our key findings are summarized in Table 1. The annual growth rate in this field of research was 9.93%, indicating a consistent rise in the number of publications over the study period.

**Table 1.** The 10 most frequently cited papers on quality of life and caregiver burden in care for older adults (2006–2023).

| Rank | Study ID [Reference] | Title | Journal | Citations | DOI |
|---|---|---|---|---|---|
| 1 | Belasco A., 2006 [25] | Quality of Life of Family Caregivers of Elderly Patients on Hemodialysis and Peritoneal Dialysis | *American Journal of Kidney Diseases* | 148 | 10.1053/j.ajkd.2006.08.017 |
| 2 | Martinez-Martin P., 2008 [43] | Burden, Perceived Health Status, and Mood among Caregivers of Parkinson's Disease Patients | *Movement Disorders* | 131 | 10.1002/mds.22106 |
| 3 | Laks J., 2016 [39] | Caregiving for Patients with Alzheimer's Disease or Dementia and Its Association with Psychiatric and Clinical Comorbidities and Other Health Outcomes in Brazil | *International Journal of Geriatric Psychiatry* | 53 | 10.1002/gps.4309 |
| 4 | Black S.E., 2010 [26] | Canadian Alzheimer's Disease Caregiver Survey: Baby-Boomer Caregivers and Burden of Care | *International Journal of Geriatric Psychiatry* | 39 | 10.1002/gps.2421 |

**Table 1.** *Cont.*

| Rank | Study ID [Reference] | Title | Journal | Citations | DOI |
|---|---|---|---|---|---|
| 5 | Lethin C., 2017 [41] | Psychological Well-Being Over Time among Informal Caregivers Caring for Persons with Dementia Living at Home | *Aging and Mental Health* | 38 | 10.1080/13607863.2016.1211621 |
| 6 | Simonelli C., 2008 [55] | The Influence of Caregiver Burden on Sexual Intimacy and Marital Satisfaction in Couples with An Alzheimer Spouse | *International Journal of Clinical Practice* | 37 | 10.1111/j.1742-1241.2007.01506.x |
| 7 | Dionne-Odom J.N., 2020 [34] | Effects of A Telehealth Early Palliative Care Intervention for Family Caregivers of Persons with Advanced Heart Failure: The ENABLE CHF-PC Randomized Clinical Trial | *JAMA Network* | 34 | 10.1001/jamanetworkopen.2020.2583 |
| 8 | Menn P., 2012 [45] | Dementia Care in The General Practice Setting: A Cluster Randomized Trial on The Effectiveness and Cost Impact of Three Management Strategies | *Value in Health* | 29 | 10.1016/j.jval.2012.06.007 |
| 9 | Olai L., 2015 [46] | Life Situations and The Care Burden for Stroke Patients and Their Informal Caregivers in A Prospective Cohort Study | *Upsala Journal of Medical Sciences* | 24 | 10.3109/03009734.2015.1049388 |
| 10 | Andrieu S., 2007 [22] | New Assessment of Dependency in Demented Patients: Impact on The Quality of Life in Informal Caregivers | *Psychiatry and Clinical Neurosciences* | 23 | 10.1111/j.1440-1819.2007.01660.x |

We included a total of 1638 references and identified 152 unique author keywords. Twenty-seven percent of the authors were engaged in collaborative research.

### 3.2. Trend of Publication and Citation

The number of publications (*n*) (Figure 2) varied between the years, with the peak in 2022 (*n* = 7) (Figure 2A). The number of citations per article also varied over the years, with the maximum of 8.22 in 2006 (Figure 2B).

Nine core journals, constituting a substantial portion of the total articles published on the theme of the study, were identified using Bradford's law (Figure 3). *Archives of Gerontology and Geriatrics* and *BMJ Open* published three articles each, accounting for 13.6% of the total publications within the study period. *Gerontologists* received the highest number of citations. Among the top 10 most relevant lists, the *Journal of the American Geriatrics Society* ranked first in the top 10 most cited journals (n = 73); it also had the highest impact factor (IF = 5.7). The journal with the highest IF in the cited source list was the *Journal of the American Medical Association* (IF = 120.7), followed by the *Neurology* (IF = 10.1) and *Stroke* (IF = 8.4) (Table 2).

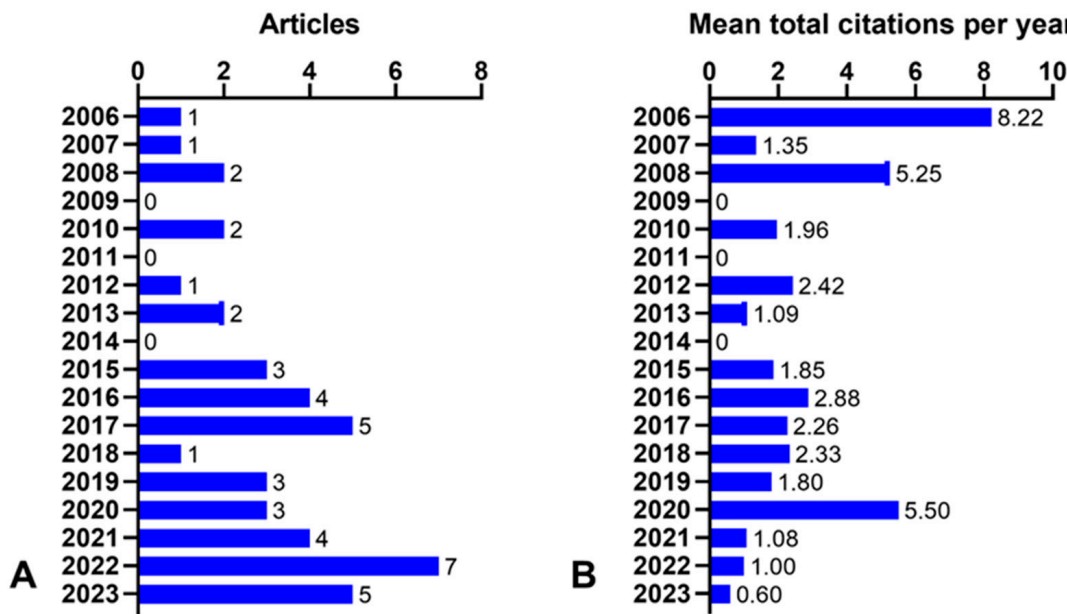

**Figure 2.** Annual trends in (**A**) the number of publications and (**B**) mean total citations per year in the field of quality of life and caregiver burden among those providing care to older adults (2006–2023).

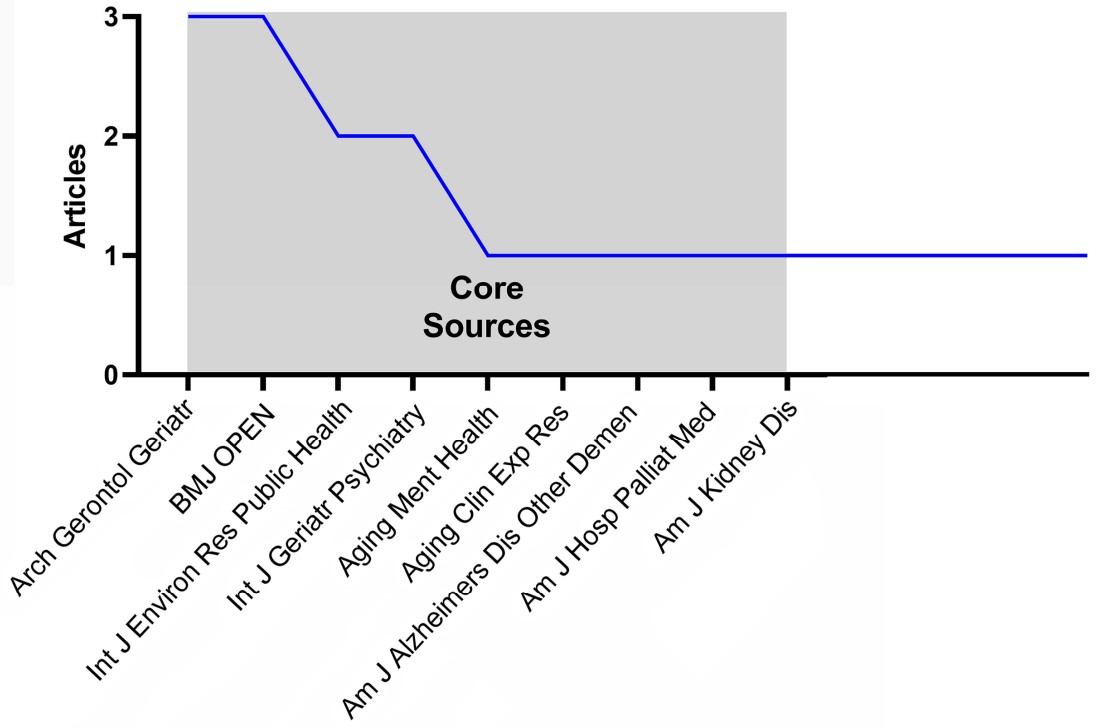

**Figure 3.** Nine core journals and the number of papers on the study topic published per journal in 2006–2023.

**Table 2.** The 10 journals with the most citations on the study topic in 2006–2023.

| Sources | Articles | IF | JCR Category (Quartile) |
|---|---|---|---|
| *Gerontologist* | 73 | 5.7 | Gerontology-SCIE (Q1); Geriatrics and Gerontology-SCIE (NA) |
| *Journal of the American Geriatrics Society* | 33 | 6.3 | Geriatrics and Gerontology-SCIE (Q1); Gerontology-SSCI (Q1) |

**Table 2.** *Cont.*

| Sources | Articles | IF | JCR Category (Quartile) |
|---|---|---|---|
| *Aging & Mental Health* | 27 | 3.4 | Geriatrics and Gerontology-SCIE (Q3); Gerontology-SSCI (Q2) |
| *International Psychogeriatrics* | 24 | 7.0 | Geriatrics and Gerontology-SCIE (Q1); Gerontology-SSCI (Q1) |
| *Journals of gerontology. Series B, Psychological sciences and social sciences* | 24 | 6.2 | Geriatrics and Gerontology-SCIE (Q1); Gerontology-SSCI (Q1) |
| *Stroke* | 24 | 8.4 | Clinical Neurology-SCIE (Q1); Peripheral Vascular Disease-SCIE (Q1) |
| *International Journal of Geriatric Psychiatry* | 23 | 4.0 | Geriatrics and Gerontology-SCIE (Q2); Gerontology-SSCI (Q2) |
| *Quality of Life Research* | 20 | 3.5 | Health Care Sciences and Services-SCIE (Q2); Health Policy and Services-SSCI (Q2) |
| *Journal of the American Medical Association* | 17 | 120.7 | Medicine, General and Internal-SCIE (Q1) |
| *Neurology* | 16 | 10.1 | Clinical Neurology-SCIE (Q1) |

*3.3. Most Productive Institutions, Authors, Countries, and Their Collaboration Network*

Kaohsiung Medical University (Taiwan) published 11 articles, accounting for 25% of the total research output on the topic (Figure 4A). The authors with the highest numbers of published papers are listed in Figure 4B. Figure 5 summarizes the interplay between the cited references, authors, and author keywords.

Spain and the USA were the leaders in scientific production, contributing 57 and 34 publications, respectively, followed by China with 28 articles, France with 19, and the UK with 15 papers (Table 3). Single-country publications comprised 80.0% and 71.4% in Spain and the USA, respectively. In contrast, Brazil, Canada, Italy, and Singapore had 100% of the research output produced in cooperation with colleagues from other countries. The strongest collaboration was observed between European countries (Figure 6).

**Table 3.** Ten countries with the most publications in the field of quality of life and caregiver burden for caregivers to older adults (2006 to 2023).

| Country | Publications |
|---|---|
| Spain | 57 |
| USA | 34 |
| China | 28 |
| France | 19 |
| UK | 15 |
| Germany | 14 |
| Canada | 11 |
| Brazil | 10 |
| Singapore | 9 |
| Sweden | 9 |

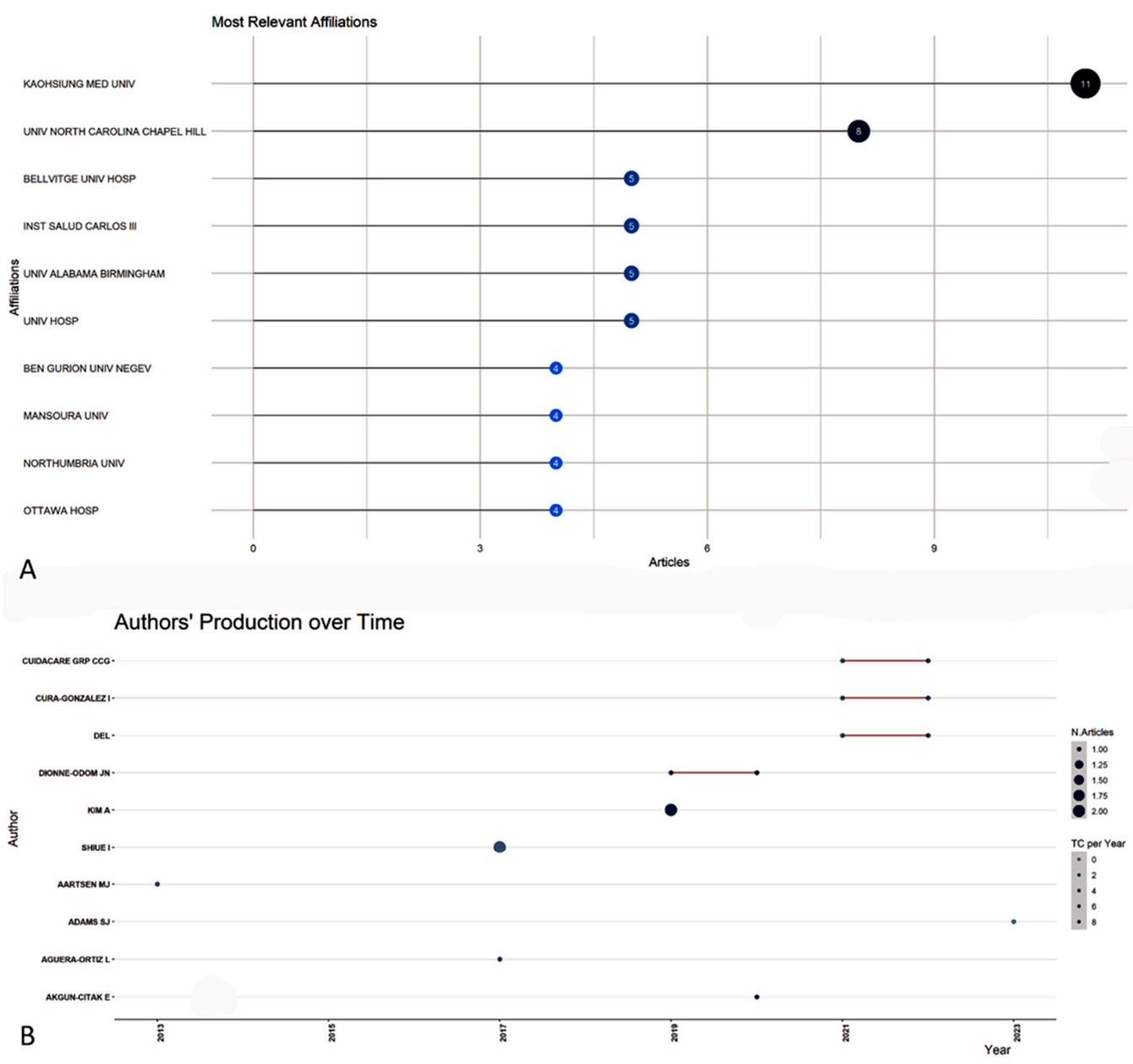

**Figure 4.** (**A**) Leading institutions, authors, countries, and their collaborative network. (**B**) Top ten contributing authors and their publication output in 2006–2023.

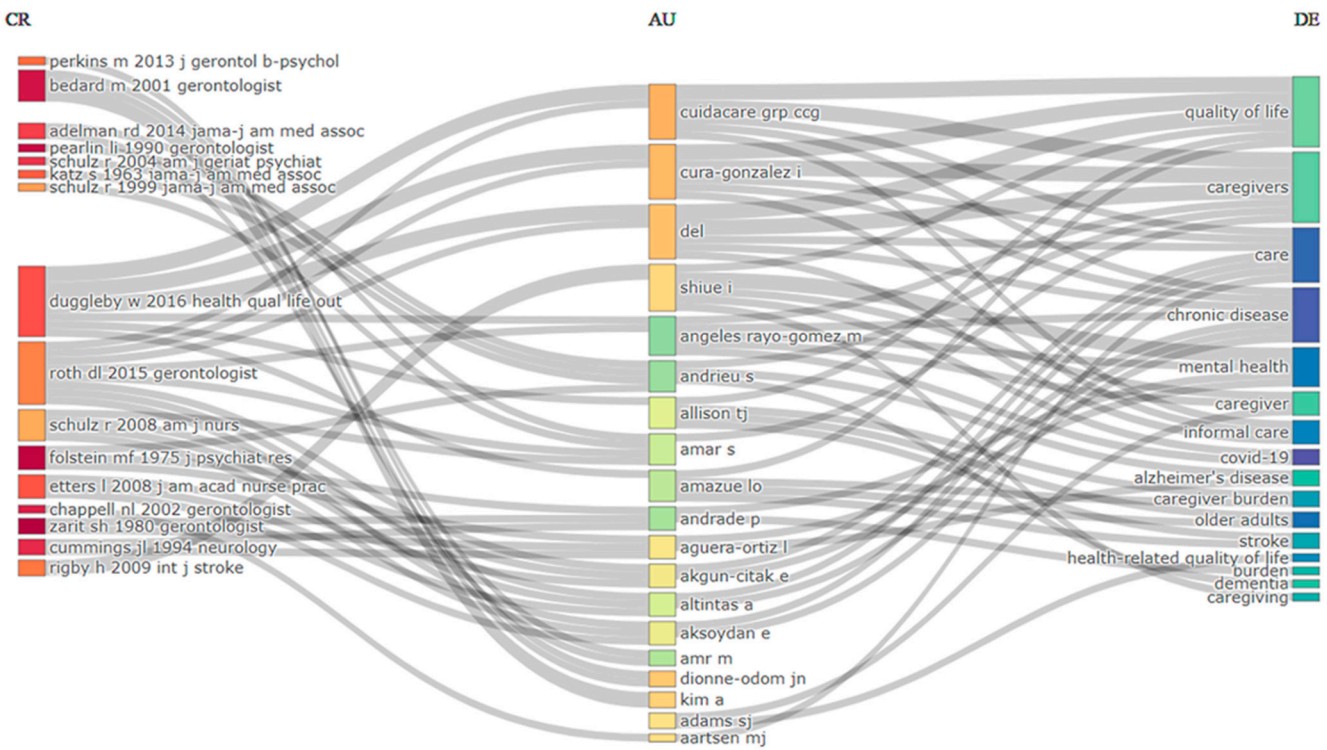

**Figure 5.** Interactions between cited references, authors [64–79], and author keywords on the topic of the study (2006–2023). Abbreviations: CR for cited references, AU for authors, and DE for keywords. Similar colored boxes indicate the similarity of incoming and outgoing flow counts in each column.

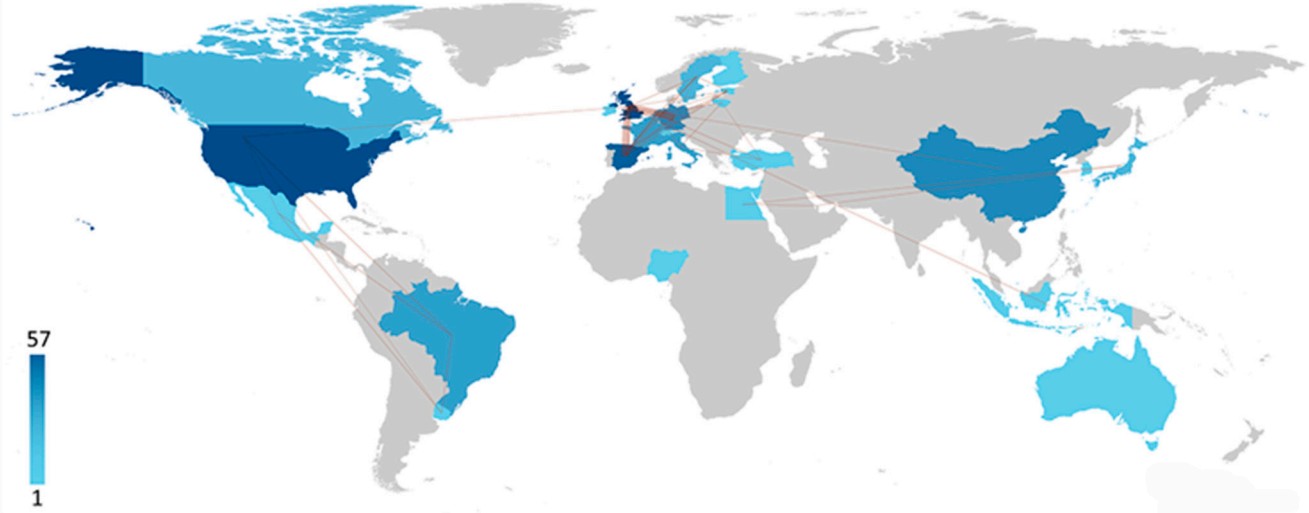

**Figure 6.** Global collaboration in the field of quality of life and caregiver burden for caregivers to older adults (2006–2023). The color reflects the number of publications per country. The strength of international cooperation is represented by the thickness of the connecting arrows.

### 3.4. Co-Occurrence, Hotspots, and Emerging Keywords

The results of the most frequently occurring author keywords are presented in Figure 7A. Keywords linked to quality of life and caregivers displayed an upward trajectory, with 16 and 7 instances in 2023, respectively (Figure 7B).

A timeline analysis of significant keywords highlighted that "Care" reached its peak of citations in 2017. Keywords such as burden, health, and impact also indicated an increasing interest in comprehending the burden and health of caregivers (Figure 8).

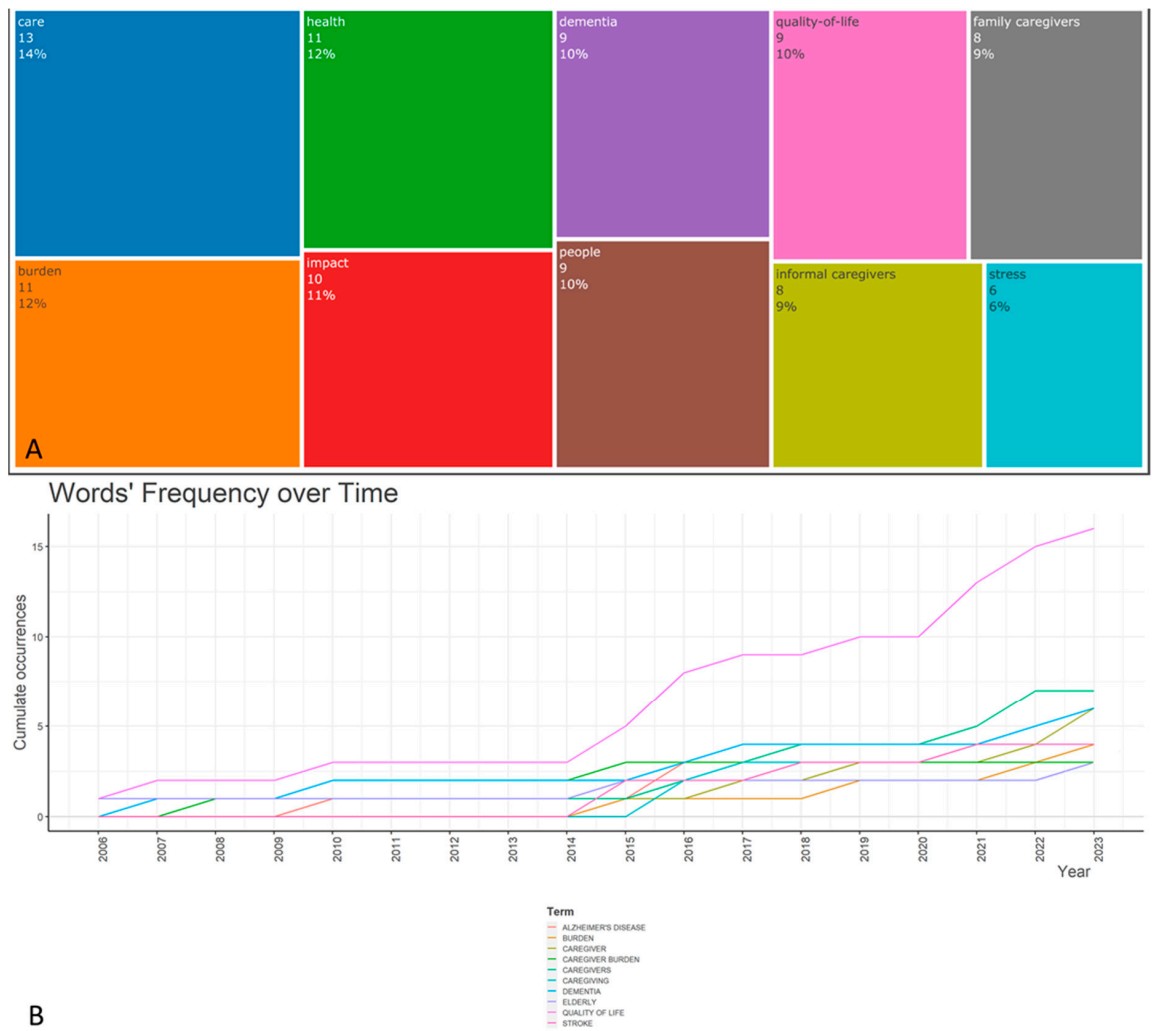

**Figure 7.** (**A**) TreeMap and (**B**) scatterplot illustrating the top ten author keywords in the field of quality of life and caregiver burden for caregivers to older adults (2006–2023).

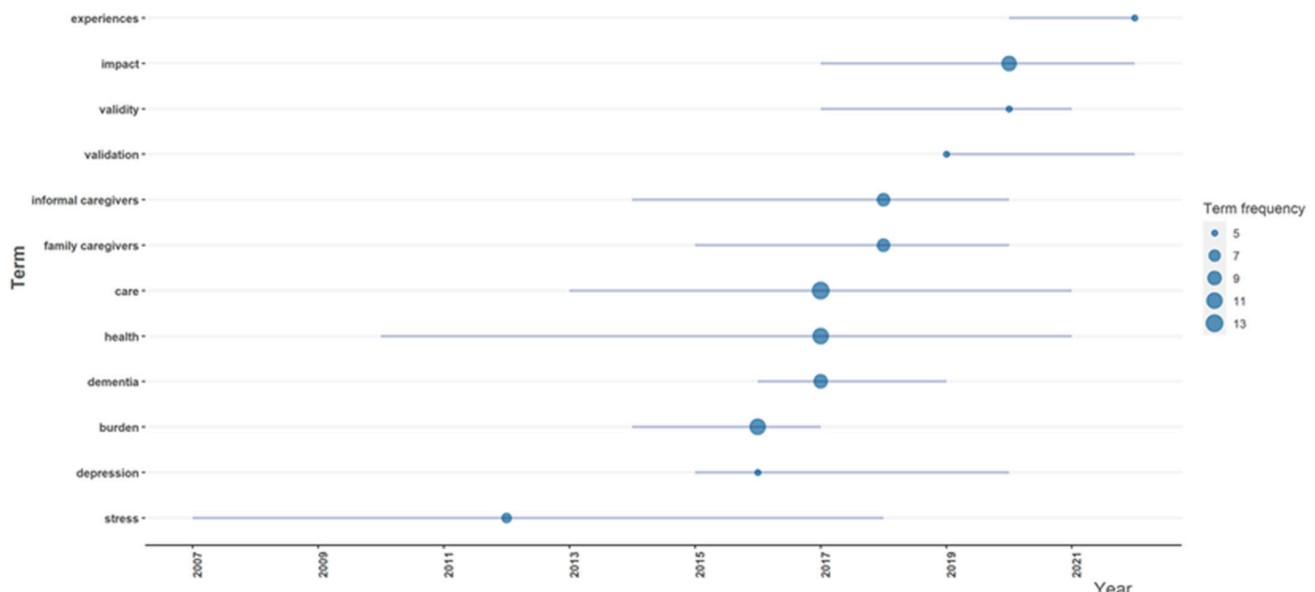

**Figure 8.** The chronological representation of trending topics. Each bubble signifies the peak frequency of use for a particular topic, while the line indicates the years in which it was employed.

## 4. Discussion

This bibliometric analysis serves to illustrate the scope and characteristics of the scientific literature on the burden and quality of life among caregivers for older adults. The timeframe for our analysis was determined based on the earliest article available in the WOS-CC database, from 2006 to 2023. Several findings deserve attention, such as (1) the patterns in how research is published and cited, (2) the most common words used by authors, (3) how authors, institutions, and countries work together on research, and (4) how these important discoveries can impact policymakers and healthcare providers.

The analysis revealed a substantial increase in research articles focusing on the challenges faced by caregivers and their quality of life in the past ten years [80,81]. This upward trend indicates a growing recognition of the critical importance of understanding caregiver burden and quality of life, as well as their implications for those who care for older adults. The growing volume of published research highlights the pressing need to tackle worldwide caregiving issues, particularly those associated with the stress on caregivers and their overall quality of life. It also underscores the importance of exploring innovative approaches to address these challenges efficiently [82].

In a wider perspective, there has been a clear rise in the number of published studies exploring the difficulties and wellbeing of individuals who care for older adults. This trend is directly connected to the worldwide phenomenon of an aging population [83]. A significant proportion of impactful publications primarily feature quantitative research initiatives designed to evaluate caregiver burden and quality of life via the administration of standardized instruments and surveys. As an illustrative example, Belasco et al. [25], Martinez-Martin et al. [43], and Laks et al. [39] conducted research using structured questionnaires to reveal that participants displayed a decrease in their quality-of-life scores while concurrently experiencing a substantial increase in their burden scores. The top 10 studies, in terms of average annual citations, covered a range of research types. These included (1) cross-sectional studies aimed at quantifying caregiver burden and quality of life, (2) mixed-methods studies combining questionnaires with semi-structured interviews to gain a more profound understanding of the factors contributing to caregiver burden, and (3) intervention studies focused on alleviating psychosocial stress among caregivers. This trend suggests that researchers globally are progressively involved in investigating the complex challenges encountered by individuals providing care for older adults and are dedicated to addressing or enhancing these matters [84]. This emphasizes the importance of research on the emotional wellbeing, vitality, and mental health of caregivers. The possibility of exploring caregiver burden and quality of life as essential components within a multidisciplinary healthcare framework is an emerging area in healthcare research.

The prominent journals in this field included a mix of gerontology and aging journals, general medical journals, and public-health- and kidney-disease-related journals. Additionally, all of the journals in the core group were highly influential. The choice of high-impact peer-reviewed journals is vital to ensure the credibility of research findings, thereby ensuring the quality of evidence [85]. This is important because many policies and healthcare providers rely on high-quality evidence in decision-making [86]. Authors take various factors into account when deciding which journals to submit their work to. These factors include the journal's reputation, its impact factor, whether it offers open access, whether it is indexed in databases, the speed of the peer-review process, the likelihood of acceptance, and the cost of publication [87].

Publications on the studied topic predominantly originate from high-income nations, including Spain and the United States, which corresponds to the findings of Bin Suliman et al. [88]. The majority of the most influential institutions in this field were from Taiwan and the United States, with some contributions from institutions in Europe and Asia. This implies that a substantial portion of the research has been conducted in high-income countries. These findings are consistent with broader patterns previously identified in caregiver studies [5]. Caregiving may exert an adverse impact on mental health globally, with a more pronounced effect observed in high-income nations [89]. Moreover, an increase in the

number of older adults requiring care in developing countries may further complicate the situation [83]. At the same time, our study revealed a noticeable underrepresentation of authors from low-income nations. This may signify that these regions are grappling with more pronounced and substantial challenges due to deficiencies in their support services [90]. This study highlighted a noticeable underrepresentation of studies from low-income countries in caregiving research, emphasizing the need for a more detailed discussion on the associated challenges. Factors contributing to this imbalance include limited research funding, inadequate institutional support, and constrained educational opportunities in low-income settings. Language barriers and restricted access to technological resources further hinder active participation in research. Additionally, the impact of geopolitical factors on collaboration needs careful consideration.

In our bibliometric analysis, Spain emerged as a prominent contributor to the field, with several universities actively engaging in research, representing 21.9% of the total number of publications. However, Kaohsiung Medical University, located in Taiwan, exhibited the highest concentration of research output among academic institutions, contributing to 25% of the publications. It is crucial to distinguish that the percentage attributed to Spain reflects the collective contribution of multiple universities, each contributing to the overall research landscape. On the other hand, the identification of Kaohsiung Medical University in Taiwan underscores a notable concentration of research productivity within a specific institution.

Our study on international research collaborations revealed interesting trends. While countries like Brazil, Canada, Italy, and Singapore actively engage in collaborative research, European nations stand out as key players. European research entities demonstrate robust interconnections, highlighting substantial international collaboration in this region. This cooperative approach raises questions about the benefits of such efforts, as they facilitate diverse perspectives, expertise, and resources, leading to more comprehensive research outcomes. Additionally, collaboration enhances global visibility for researchers and institutions, streamlining access to global networks and funding. Variations in healthcare systems, cultural norms, family structures, and caregiving methods between nations contribute to distinct challenges, possibly explaining the disparities in international collaboration in this research field [8,91]. To address these global healthcare challenges effectively, active engagement in collaborative endeavors is imperative for public health professionals.

Within the European Union, collaborative research initiatives have become a cornerstone, fostering interdisciplinary partnerships and knowledge exchange that significantly benefit both the local and global contexts of caregiving research. The robust collaboration within the European Union contributes to an enriched understanding of caregiving dynamics, not only within the region but with implications that extend globally. By cultivating a shared pool of expertise and resources, collaborative efforts within the EU elevate the quality of research and inform best practices in caregiving. Furthermore, these collaborations act as catalysts for innovation and can potentially serve as models for similar initiatives beyond Europe. Through this interconnected approach, the European Union plays a pivotal role in advancing caregiving research on a global scale, with its collaborative endeavors serving as a beacon for promoting best practices and informed caregiving strategies worldwide

This study explored various complex concepts related to caregivers for older adults and identified key terms used by authors. These terms included "burden", "caregiver", "care", "health", "quality of life", "stress", "dementia", and "impact". They included the caregiver's role (1), the challenges they face (2), and the needs of both older adults and caregivers (3), including care and health. Notably, research increasingly emphasizes understanding caregivers' quality of life, which depends on their caregiving duties and the care recipients' health, showing an inverse relationship with caregiver burden. Researchers and healthcare professionals now focus on comprehending the factors contributing to increased caregiver burden and reduced quality of life, to develop preventive strategies.

Our study highlights diverse topics within this field: (1) exploring caregivers' experiences, (2) identifying side effects and their determinants, and (3) developing interventions

for informal caregivers and older adults. Research on stroke caregivers is increasingly vital due to their growing role in healthcare. Our findings can guide future research, shape research directions, and inform policy. Moreover, this can motivate researchers in low- and middle-income countries to assess the burden and quality of life among caregivers for older adults.

*Limitations of the Study*

While our bibliometric analysis provides valuable insights into the landscape of research on the quality of life and caregiver burden among those caring for older adults, it is important to acknowledge certain limitations that may influence the interpretation of our findings.

Database limitations: Our analysis was based on data retrieved from the WOS-CC database. The exclusion of other databases may have resulted in a partial representation of the global literature on the subject. Different databases may contain unique sets of publications that could contribute to a more comprehensive overview.

Language bias: The requirement for articles to be authored in English may have introduced a language bias, potentially excluding relevant studies published in other languages. This limitation may impact the inclusivity and diversity of the literature considered in our analysis.

Publication trends: Our study focused on publications available up to the current year (2023). The dynamic nature of research may lead to ongoing developments beyond this timeframe. The choice of a fixed endpoint may not capture emerging trends or shifts in the research landscape.

Citation analysis limitations: While we conducted citation analysis to identify impactful studies, it is essential to recognize that citation counts alone may not be indicative of the quality or significance of a publication. Variations in citation practices between disciplines, along with other factors, can influence citation metrics.

Subjectivity in keyword selection: The identification of key terms for analysis involves a degree of subjectivity. Our selection of keywords may have influenced the scope of the analysis, potentially excluding relevant publications that employ different terminology.

Geographic representation: Our analysis revealed a concentration of contributions from high-income nations, particularly Spain, the United States, and China. The limited representation of low-income countries and regions may affect the generalizability of our findings to diverse global contexts.

Interpretation of collaborations: While we identified collaborative patterns between countries and institutions, the qualitative aspects of these collaborations were not extensively explored. The nature and impact of collaborative efforts could provide additional context to our analysis.

Despite these limitations, our bibliometric analysis serves as a valuable contribution to understanding the scholarly landscape related to caregivers for older adults. Researchers should consider these limitations when interpreting the results, and they may build upon our work to further explore the complexities of this important field.

## 5. Conclusions

This bibliometric analysis summarizes the published research on caregivers for older adults, highlighting significant publications, authors, and international journals, while emphasizing the urgency of addressing global caregiving challenges. Collaborative research, particularly in Europe, underscores the benefits of international partnerships for innovative outcomes. However, the limited collaboration in some regions and underrepresentation of authors from low-income countries reveal challenges due to inadequate support services. Promoting international collaboration is crucial to address global healthcare challenges. Our analysis also emphasizes the growing interest in understanding caregivers' quality of life and the wide range of topics covered in publications. These findings provide insights

for future research and policy, particularly regarding the assessment of caregivers' burden and quality of life.

Aligning caregiving practices with sustainable healthcare paradigms necessitates a concerted effort towards equitable support services and fostering global collaboration. By acknowledging the multidimensional nature of caregiving challenges and promoting international cooperation, strides can be made towards sustainable caregiving practices that ensure the wellbeing of both caregivers and the aging population, thus safeguarding the sustainability of healthcare systems.

**Author Contributions:** Conceptualization, A.Z. (Aliya Zhylkybekova) and A.T.; data curation, A.Z. (Aliya Zhylkybekova), G.K.K. and A.M.G.; formal analysis, A.Z. (Aliya Zhylkybekova); investigation, A.Z. (Aliya Zhylkybekova); methodology, A.Z. (Aliya Zhylkybekova) and A.T.; project administration, G.K.K.; resources, A.Z. (Afshin Zare), N.M.M., A.A.K., S.B. and N.A.; software, A.Z. (Aliya Zhylkybekova), N.A. and A.T.; supervision, G.K.K., A.M.G. and N.G.; validation, A.Z. (Aliya Zhylkybekova); writing—original draft, A.Z. (Aliya Zhylkybekova) and A.T.; writing—review and editing, G.K.K., A.Z. (Afshin Zare), N.M.M., A.A.K., S.B., N.A., A.M.G. and N.G. All authors have read and agreed to the published version of the manuscript.

**Funding:** This research received no external funding.

**Institutional Review Board Statement:** Not applicable.

**Informed Consent Statement:** Not applicable.

**Data Availability Statement:** Data are available upon request due to ethical restrictions.

**Conflicts of Interest:** Authors Afshin Zare and Amin Tamadon are employed by PerciaVista R&D Co. The remaining authors declare that the research was conducted in the absence of any commercial or financial relationships that could be construed as a potential conflict of interest.

## Appendix A

Detailed PICO framework for the search strategy:

| | |
|---|---|
| Population | Informal caregivers for older adults. Definition of "older adults": individuals aged 65 and above |
| Intervention | Various caregiving approaches, strategies, or support mechanisms |
| Comparison | The comparison may involve different types of caregiving interventions, the absence of a specific intervention, or a comparison between different caregiver populations |
| Outcome | Primary outcomes include the quality of life of caregivers and the burden experienced by caregivers while providing care to older adults |

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
