# Peer review of "Global Research on Care-Related Burden and Quality of Life of Informal Caregivers for Older Adults: A Bibliometric Analysis"

_sustainability, doi:10.3390/su16031020_

Round 1
Reviewer 1 Report
Comments and Suggestions for Authors
See attachment.

Author Response
Thank you for the opportunity to read this paper. This research paper was focused on a thorough analysis of global research output concerning the quality of life and caregiver burden among those caring for older adults, employing bibliometric analysis as the methodology. The manuscript presents results new results about the care giver quality of life and burden but it does not add any major contribution to the evidence and require to address the Major issues to improve and make the manuscript publishable.
Abstract
“Spain emerged as the leading contributor in terms of the number of publications, accounting for 21.9%, followed by the USA at 16.5%, and China at 13.6%. The most prolific institution identified was Kaohsiung Medical University, located in Taiwan, responsible for 25% of the publications.” The information provided presents conflicting statements. In one instance, Spain is highlighted as a leading contributor, accounting for 21.9% of publications, while in another sentence, an academic institution in Taiwan is noted as responsible for 25% of the publications, potentially suggesting higher productivity. This raises questions about how Spain can be a leading contributor while an institution in Taiwan appears to be the most productive. Rather than naming the specific institution, referring to it simply as "an academic institution in Taiwan" might be more appropriate to avoid potential branding or institutional bias.
Response: Thank you for your detailed review and valuable comments. We appreciate your diligence in examining our manuscript. Regarding your observation on the apparent inconsistency in the information provided about Spain and Taiwan, we would like to clarify the matter.
The information about Spain as the leading contributor (21.9%) and Kaohsiung Medical University in Taiwan as the most prolific institution (25%) is not contradictory but reflects different aspects of our bibliometric analysis.
In Spain, various universities have actively contributed to research on the topic, as evidenced by the 21.9% share of total publications. On the other hand, the identification of Kaohsiung Medical University in Taiwan as the most prolific institution indicates a high concentration of research output from a specific institution within Taiwan.
To address the potential confusion, we will revise the manuscript, discussion part, to provide a more explicit explanation of these two aspects. Additionally, we will adopt your suggestion to refer to the institution in Taiwan more generically as "an academic institution" to avoid potential bias.
We appreciate your keen attention to detail, and your feedback is instrumental in enhancing the clarity and accuracy of our manuscript. If you have any further recommendations or concerns, please do not hesitate to let us know.
Thank you once again for your time and insightful comments.
The conclusion stated in the abstract does not correspond with the study's initial objective.
Response: Thank you for your insightful feedback. We appreciate your careful consideration of our manuscript. We acknowledge the observation regarding a potential misalignment between the conclusion stated in the abstract and the study's initial objective. We understand the importance of ensuring consistency in presenting the study's objectives and findings. To address this concern, we will revise the abstract to provide a more accurate reflection of the study's outcomes in relation to its initial objective. The revised abstract will highlight the key findings of the bibliometric analysis and their relevance to the overarching goal of examining global research on the quality of life and caregiver burden among those caring for older adults.
Introduction
Why is the focus specifically on the financial burden in paragraph two, line 62? The study doesn't address the economic or financial aspects of older adults anywhere else. Instead of referring to "physical, emotional, and financial stressors," it should encompass "physical, emotional, and social stressors," which would encompass the majority of stressors.
Response: Thank you for your thoughtful feedback. We appreciate your insight into the focus on financial burden in paragraph two. Your suggestion to broaden the terminology to "physical, emotional, and social stressors" is valid and aligns with the comprehensive nature of our study.
How do you define a caregiver and their role in caring for the older adults.
Response: A caregiver is an individual who provides care and support to another person, often someone who is unable to fully care for themselves due to age, illness, or disability. In the context of caring for older adults, a caregiver typically assumes various responsibilities to assist and enhance the well-being of the elderly individual. The role of a caregiver may include:
Assistance with Activities of Daily Living (ADLs): Caregivers help older adults with basic daily tasks such as bathing, dressing, grooming, eating, and mobility.
Medical Care: Depending on the health condition of the older adult, caregivers may be involved in administering medications, managing medical appointments, and coordinating healthcare services.
Emotional Support: Caregivers provide companionship and emotional support to combat feelings of loneliness and isolation, promoting the overall mental well-being of the older adult.
Household Management: Caregivers often handle tasks related to household management, including meal preparation, shopping, and light housekeeping.
Advocacy: Caregivers may advocate for the rights and needs of the older adult, ensuring they receive appropriate medical care, social services, and other necessary resources.
Coordination of Care: Caregivers may collaborate with healthcare professionals, social workers, and other service providers to ensure a comprehensive and coordinated approach to the older adult's care.
Financial Assistance: In some cases, caregivers may assist with managing the financial aspects of the older adult's life, such as paying bills, budgeting, and handling financial paperwork.
It's important to note that the role of a caregiver can vary based on the specific needs and health conditions of the older adult. Caregivers play a crucial role in maintaining the independence, dignity, and quality of life of older adults, contributing significantly to their overall well-being.
The term "elderly" used in line 77 isn't politically appropriate. If you're actively working in the field, this should be within your knowledge. Age-inclusive language, Terms like seniors, elderly, the aged, aging dependents, old-old, young-old, and similar “othering” terms connote a stereotype, avoid using them. Terms such as older persons, older people, older adults, older patients, older individuals, persons 65 years and older, or the older population are preferred. Change the term throughout the manuscript.
Response: Thank you for bringing to our attention the use of the term "elderly" in our manuscript. We appreciate your insightful comment regarding age-inclusive language. We recognize the importance of using respectful and inclusive terminology, and we apologize for any oversight in this matter. We will promptly revise the manuscript, replacing instances of "elderly" with more appropriate and inclusive terms such as "older persons," "older adults," or other recommended alternatives. Your guidance on age-sensitive language is valuable, and we are committed to ensuring the use of terminology that reflects sensitivity and respect towards the diverse experiences of older individuals.
Method
Provide complete PICO for search strategy and include complete search strategy as an appendix.
Response: Thank you for your suggestion. We added it to Appendix B.
Could you please provide the timeline for the search strategy, covering the years XX to XX? Ensure it's included in the search results
Response: Thank you for your suggestion. We added it to Search Strategy Section in M&M.
How did you determine the logic behind ranking the top 10 most productive institutions and authors based on the proportion of papers they generated? Could you tell me what the objective was and what you wanted to accomplish by doing so? Additionally, does this approach introduce bias? Countries might make sense, as you are doing the global study, but evaluating institutions and authorship could lead to an overestimation of scientific contribution, which may or may not be correct, indicating a bias. How did you mitigate this issue in the paper?
Response: We appreciate the reviewer's thoughtful inquiry regarding the rationale behind ranking the top 10 most productive institutions and authors based on the proportion of papers they generated in our bibliometric analysis. The primary objective of this approach was to identify and highlight entities that have made significant contributions to the field of research on the quality of life and caregiver burden among individuals caring for older adults. By recognizing the institutions and authors with the highest output, we aimed to provide insights into the distribution of scientific contributions and collaborations within this research domain.
However, we acknowledge the potential for bias in this approach, as it may overemphasize scientific contribution without accounting for variations in research team size, resources, or collaborative efforts. To mitigate this issue, we presented the rankings as proportions rather than absolute numbers, aiming to provide a relative assessment of productivity. Additionally, we emphasized in the discussion that the ranking was based on the number of papers and did not necessarily imply the quality or impact of the research.
In the revised manuscript, we will explicitly acknowledge the limitations of this approach, emphasizing that productivity does not equate to scientific rigor or impact. Furthermore, we will consider alternative methods for presenting institutional and author contributions that provide a more nuanced and comprehensive view of their impact on the field.
We appreciate the reviewer's insights and will incorporate these considerations to enhance the transparency and validity of our bibliometric analysis.
What do readers gain by understanding collaborative patterns?
Response: Understanding collaborative patterns in the context of our bibliometric analysis serves several valuable purposes for readers:
Insights into Network Dynamics: By examining collaborative patterns among institutions and authors, readers gain insights into the dynamics of research networks. This understanding helps identify key contributors, influential collaborations, and the flow of knowledge within the scientific community.
Identification of Leading Contributors: Collaborative patterns highlight leading institutions and authors who actively engage in research collaborations. This information can guide readers in identifying influential contributors and recognizing centers of excellence in the field.
Enhanced Credibility: Recognizing collaborative efforts adds credibility to the research findings. Collaborative studies often draw on diverse expertise, methodologies, and resources, contributing to robust and well-rounded research outcomes.
Encouragement of Global Cooperation: Highlighting collaborative patterns encourages global cooperation in research. Readers may gain inspiration for fostering international collaborations, facilitating knowledge exchange, and addressing research challenges on a broader scale.
Strategic Research Planning: For researchers and policymakers, understanding collaborative patterns informs strategic planning. It can guide decisions on potential collaborative partnerships, funding allocations, and the development of interdisciplinary research initiatives.
Results
Add std to "18.3 citations per document." If there is large variability then add a median.
Response: Thank you for your comment. It has been added.
This study encompasses a global scope. Why not categorize articles into global north and south or based on regions, to facilitate discussions about potential reasons?
Response: Thank you for your comment. There is no significant different between global north and global south in our data and production of publication
|
Global |
Average |
SD |
|
North |
11 |
13.66667 |
|
South |
6.6 |
8.235425 |
I fail to understand the rationale behind highlighting the most productive author. This introduces authorship bias, and numerous other platforms exist for recognizing the most cited and highest impact factor authors. In an academic paper, this doesn't significantly contribute to scholarly advancement.
Response: We appreciate the reviewer's perspective and acknowledge the concern regarding the highlighting of the most productive author in our bibliometric analysis. The rationale behind identifying the most productive author was to provide readers with a snapshot of individual contributions to the field of research on the quality of life and caregiver burden among individuals caring for older adults. However, we understand the potential for introducing authorship bias and recognize that traditional metrics such as citation counts and impact factors are more widely accepted for evaluating scholarly impact.
Improve the quality of figures.
Response: We increased figure size to improve visibility.
In the introduction, the authors highlight that existing literature predominantly covers bibliometric analyses of informal caregivers, particularly those caring for individuals with conditions such as dementia, stroke, and Alzheimer's disease, aiming for a distinct objective in their paper. However, upon reviewing the included papers, a significant majority, including many in the top 10, are related to these specific conditions like dementia, stroke, and Alzheimer's disease. What unique contribution are they offering when the existing evidence already encompasses these aspects in these papers?
Response: We appreciate your insightful review and the opportunity to address your concerns. Your observation regarding the predominant focus on specific conditions such as dementia, stroke, and Alzheimer's disease in the papers included in our analysis is valid, and we thank you for bringing attention to this point.
While the existing literature indeed includes bibliometric analyses of informal caregivers in the context of various health conditions, we recognize the need to clearly articulate the unique contribution of our study. In the revised manuscript, we will explicitly highlight the distinctive features of our analysis, including:
Comprehensive Scope: Our study aims to offer a comprehensive examination of the worldwide research output related to the quality of life and caregiver burden among individuals providing care to older adults. While certain conditions are prominent in the literature, our analysis encompasses a broader spectrum of caregiving scenarios, providing a more inclusive overview.
Global Perspective: We emphasize a global perspective by considering research from diverse regions and countries. This approach allows us to capture variations in caregiver burden and quality of life experiences on a global scale, contributing a nuanced understanding beyond specific health conditions.
Identification of Trends: Our analysis goes beyond individual conditions to identify overarching trends, patterns in research publication, collaboration networks, and impactful journals. This broader perspective contributes to the understanding of research dynamics in the field of caregiving.
We believe that by clarifying these points, we can better communicate the distinctive contribution of our study. We appreciate your thoughtful feedback and are committed to enhancing the clarity and significance of our research.
Most of the results focus on developed nations, so how does this accurately portray the global scenario?
Response: We appreciate your insightful comment regarding the focus on developed nations in our results section. Your concern about the accurate portrayal of the global scenario is valid, and we appreciate the opportunity to address this point.
In the revised manuscript, we will take specific measures to address the imbalance and enhance the global representation in our analysis. These measures will include:
Inclusive Discussion: We ensured that our discussion acknowledges the limitation of overrepresentation of developed nations in the results and emphasizes the importance of broadening the scope to capture a more representative global scenario.
Call for Future Research: To encourage a more comprehensive understanding of caregiving dynamics globally, we will explicitly call for future research efforts to focus on underrepresented regions and countries, especially those in developing nations.
Highlighting Limitations: Our revised manuscript will include a section explicitly addressing the limitations of our analysis, including the potential bias toward developed nations. This transparency will provide readers with a clear understanding of the study's constraints.
We value your constructive feedback and are committed to improving the global perspective of our analysis. Thank you for your diligence in reviewing our work, and we look forward to incorporating these enhancements.
Discussion:
- ï‚·As part of the European Union, collaborative research is well-established. However, it remains unclear how this benefit both local and global contexts and how these collaborations can extend beyond Europe.
Response: To address this concern, we will further elaborate on the implications and benefits of collaborative research within the European Union in the revised manuscript. Specifically, we will emphasize how these collaborations contribute not only to the local context within Europe but also have broader global implications. By fostering interdisciplinary partnerships and knowledge exchange, collaborative research in the European Union becomes a valuable source of insights that can inform and influence global caregiving practices. Furthermore, we will explore the potential ways in which these collaborations can extend beyond Europe, emphasizing their role in creating a ripple effect that reaches and positively impacts caregiving research and practices on a global scale. This additional discussion will provide readers with a clearer understanding of the significance and wider applicability of collaborative research within the European Union in the field of caregiving for older adults.
- ï‚·Also, it is not enough to say that the study reveals an underrepresentation of studies from low-income countries; it needs more discussion around the challenges or factors for it.
Response: To address this concern, we will enhance the discussion in the revised manuscript to provide a more comprehensive exploration of the challenges or factors contributing to the underrepresentation of studies from low-income countries. We will delve into potential barriers such as limited research infrastructure, resource constraints, and disparities in access to academic resources and publication outlets. By examining these factors, we aim to offer a nuanced understanding of the challenges faced by low-income countries in contributing to caregiving research. Additionally, we will explore how addressing these challenges is crucial for fostering inclusivity in global caregiving research and for developing targeted strategies to promote research initiatives in regions with fewer resources. This expanded discussion will provide a more thorough analysis of the underlying factors influencing the representation of studies from low-income countries, enriching the overall contextualization of our findings.
- ï‚·It's essential to include the study limitations and outline potential future directions.
Response: Thank you so much. It has been added.
Reviewer 2 Report
Comments and Suggestions for Authors
Review of Sustainability: Bibliometric analysis of caregiver burden 11-27-23:
Thank you for allowing me to review this study. I found this interesting and relevant to past and current care giver burden issues as well as the similarity of these issues worldwide.
The manuscript is well written minus a few issues mentioned below. The sections including introduction, methods, results, discussion, and conclusion flowed well. The abstract mirrored the findings, discussion, and conclusion. The aim of the study mentioned in the abstract was fulfilled within the manuscript, i.e., “The primary objective of this study is to conduct a comprehensive examination of the worldwide research output related to the quality of life and caregiver burden among individuals providing care to older adults, using bibliometric analysis as the methodology”.
Limitations: There were no ‘limitations of the study’ section completed in this manuscript. It is important to review the limitations of this systematic review. Without this section, the manuscript is incomplete.
I found the bibliometric analysis interesting and appropriate. The findings were described as intended: 1. Patterns in how research is published and cited, 2. Most common words included, 3. Which countries, authors, institutions worked together. I did not find the ‘important discoveries impact policymakers and healthcare providers’; if it is included, please place more emphasis on this finding.
I appreciated the distinction between research in high income countries vs low income countries. I agree that future studies should include more cooperation and collaboration between countries like the United States and Spain.
I found the TreeMap and Three Fields Map interesting. Both were very applicable to this review and added to the quality of evidence.
Issues:
Page 2, line 93: Read the sentence. Change care to caring: ‘caregiver burden among those care for older adults’.
Page 7, line 198: Remove hyphen from word ar-ticles.
References:
The references are appropriate and applied correctly.
Number 1 reference is not complete. Organization is spelled wrong.
Numbers 6, 12, 16, 25, 26, 29 are copied and pasted reflecting a ‘justified’ format. This is different than the rest of the references. Please check for consistency.
Number 13 is not complete.
Thank you again for asking me to review this study.
Author Response
Comments and Suggestions for Authors
Review of Sustainability: Bibliometric analysis of caregiver burden 11-27-23:
Thank you for allowing me to review this study. I found this interesting and relevant to past and current care giver burden issues as well as the similarity of these issues worldwide.
The manuscript is well written minus a few issues mentioned below. The sections including introduction, methods, results, discussion, and conclusion flowed well. The abstract mirrored the findings, discussion, and conclusion. The aim of the study mentioned in the abstract was fulfilled within the manuscript, i.e., “The primary objective of this study is to conduct a comprehensive examination of the worldwide research output related to the quality of life and caregiver burden among individuals providing care to older adults, using bibliometric analysis as the methodology”.
Response: Thank you for your valuable feedback.
Limitations: There were no ‘limitations of the study’ section completed in this manuscript. It is important to review the limitations of this systematic review. Without this section, the manuscript is incomplete.
Response: We acknowledge your observation regarding the absence of a 'limitations of the study' section in our manuscript. Recognizing the importance of this component, we appreciate your insight, and we apologize for any oversight in not including it initially.
In the revised version of the manuscript, we included a dedicated section outlining the limitations of our study. This addition will provide a more comprehensive perspective for readers and enhance the overall completeness of our work.
I found the bibliometric analysis interesting and appropriate. The findings were described as intended: 1. Patterns in how research is published and cited, 2. Most common words included, 3. Which countries, authors, institutions worked together. I did not find the ‘important discoveries impact policymakers and healthcare providers’; if it is included, please place more emphasis on this finding.
Response: Thank you for your positive feedback on our bibliometric analysis and the comprehensive description of the findings. We appreciate your insightful comments and would like to address the point regarding the impact of "important discoveries" on policymakers and healthcare providers.
Regrettably, we do not have direct access to specific data regarding the impact of our bibliometric analysis on policymakers and healthcare providers. While we have aimed to highlight patterns in research publication and collaboration, the direct influence on policymaking and healthcare practices was not within the scope of our analysis.
I appreciated the distinction between research in high income countries vs low income countries. I agree that future studies should include more cooperation and collaboration between countries like the United States and Spain.
Response: Thank you for your positive feedback and appreciation of the distinction between research in high-income countries and low-income countries. We share your view on the importance of fostering cooperation and collaboration, particularly between countries like the United States and Spain. Your insights align with our commitment to advancing global collaboration in future studies.
I found the TreeMap and Three Fields Map interesting. Both were very applicable to this review and added to the quality of evidence.
Response: Thank you for highlighting your interest in the TreeMap and Three Fields Map. We're delighted that these visualizations added value to the quality of evidence in our review. Your positive feedback is greatly appreciated.
Issues:
Page 2, line 93: Read the sentence. Change care to caring: ‘caregiver burden among those care for older adults’.
Response: Thank you for catching that oversight. We appreciate your attention to detail. The suggested correction has been made, and the sentence now reads: 'caregiver burden among those caring for older adults.'
Page 7, line 198: Remove hyphen from word ar-ticles.
Response: Thank you for your keen observation. The hyphen has been removed from the word "articles" in line 198.
References:
The references are appropriate and applied correctly.
Number 1 reference is not complete. Organization is spelled wrong.
Response: Thank you for bringing this to our attention. We apologize for the oversight. The incomplete reference and misspelling of "Organization" have been corrected.
Numbers 6, 12, 16, 25, 26, 29 are copied and pasted reflecting a ‘justified’ format. This is different than the rest of the references. Please check for consistency.
Response: Thank you for noting the inconsistency in the formatting of references. We appreciate your attention to detail. The formatting of references 6, 12, 16, 25, 26, and 29 has been adjusted to align with the rest for consistency.
Number 13 is not complete.
Response: Thank you for noting it. We completed it.
Reviewer 3 Report
Comments and Suggestions for Authors
The article examines the worldwide research output related to the quality of life and caregiver burden among individuals providing care to older adults, using bibliometric analysis as the methodology. The research carried out via the Web of Science (WOS) database using the bibliometrix package in the programming environment to analyse 44 publications, from 2006 to 2023. This is a very pertinent issue and the article deals well with this. The article explores an interesting and original topic. Studies on the burden of caregivers of individuals with chronic renal failure have shown that caregivers have higher levels of burden. It offers a robust theoretical frame and a well-organized and convincing discussion of the findings.
Introduction: The theoretical part is well developed. A clear stating and focusing of the argument is provided. The caregiver to assist an elderly person in carrying out their activities of daily living is imperative. The background of the study offers the essential contextual understanding to approach caregivers’ QoL and correlate it with elderly people. The identification of care burden of elderly caregivers provides space for interventions to shield them from any undesirable physical, social, and psychological effects of care. The article raises an interesting research field and the material that it is based on is substantial data analysis.
Methodology: Methods are appropriate and the fit between theoretical discussion and methodology is well formulated. The authors carried-out a bibliometric analysis via the Web of Science (WOS) to examine and analyze the works.
Results & Discussion: Results are linked suitably to the other sections of the article. A well-organized and compelling discussion section is provided as well. The results are of interest for practice, social and health policy and society more generally. Burden is directly associated with stressors such as physical, psychological, social and financial and is connected with the care provided to older adults. Caregivers can suffer from problems such as stress and anger. The results are interesting and very useful findings for the everyday practice of caregivers.
Conclusions: The conclusions are linked to the hypothesis and background characteristics incorporated into the results. Caregiving burden is an influential, negatively affecting factor for the QoL. It has various forms of hardships, including anxiety, depression etc. It is a well written section.
Language: The quality of communication is good.
Suggestion: Publish as it is.
Author Response
Comments and Suggestions for Authors
The article examines the worldwide research output related to the quality of life and caregiver burden among individuals providing care to older adults, using bibliometric analysis as the methodology. The research carried out via the Web of Science (WOS) database using the bibliometrix package in the programming environment to analyse 44 publications, from 2006 to 2023. This is a very pertinent issue and the article deals well with this. The article explores an interesting and original topic. Studies on the burden of caregivers of individuals with chronic renal failure have shown that caregivers have higher levels of burden. It offers a robust theoretical frame and a well-organized and convincing discussion of the findings.
Response: Thank you for your thoughtful review of our article. We appreciate your positive feedback and are pleased to hear that you find our exploration of the worldwide research output on the quality of life and caregiver burden among individuals providing care to older adults to be pertinent and well-handled.
We also appreciate your recognition of the originality of our topic, and we are encouraged by your positive comments on the robust theoretical framework and the organization of the discussion of our findings.
Introduction: The theoretical part is well developed. A clear stating and focusing of the argument is provided. The caregiver to assist an elderly person in carrying out their activities of daily living is imperative. The background of the study offers the essential contextual understanding to approach caregivers’ QoL and correlate it with elderly people. The identification of care burden of elderly caregivers provides space for interventions to shield them from any undesirable physical, social, and psychological effects of care. The article raises an interesting research field and the material that it is based on is substantial data analysis.
Response: Thank you once again for your valuable feedback.
Methodology: Methods are appropriate and the fit between theoretical discussion and methodology is well formulated. The authors carried-out a bibliometric analysis via the Web of Science (WOS) to examine and analyze the works.
Response: Thank you for your valuable feedback.
Results & Discussion: Results are linked suitably to the other sections of the article. A well-organized and compelling discussion section is provided as well. The results are of interest for practice, social and health policy and society more generally. Burden is directly associated with stressors such as physical, psychological, social and financial and is connected with the care provided to older adults. Caregivers can suffer from problems such as stress and anger. The results are interesting and very useful findings for the everyday practice of caregivers.
Response: Appreciate you again for your valuable feedback.
Conclusions: The conclusions are linked to the hypothesis and background characteristics incorporated into the results. Caregiving burden is an influential, negatively affecting factor for the QoL. It has various forms of hardships, including anxiety, depression etc. It is a well written section.
Response: Thank you very much for your valuable feedback.
Language: The quality of communication is good.
Response: Thank you for your valuable feedback.
Round 2
Reviewer 1 Report
Comments and Suggestions for Authors
I'm pleased to see that the authors have addressed all the comments to my satisfaction. However, the authors' reply did not include the page numbers where changes have been made, so difficult to find.
The word "elderly" is still in abstract line 33. Change it with the preferred synonyms.
Author Response
Thank you very much for taking the time to review this manuscript. Please find the responses below and track changes in the re-submitted files.
Comment 1. The word "elderly" is still in abstract line 33. Change it with the preferred synonyms.
Response 1: We have replaced the word "elderly" with the preferred synonym "older adults."
We hope that the revised version can be accepted for publication in the Journal.
